# The Study on the Ballast Water Management of Mailiao Exclusive Industrial Harbor in Taiwan

**Hao-Nan Hung** and **Ray-Yeng Yang** *

Department of Hydraulic and Ocean Engineering, National Cheng Kung University, Tainan 701401, Taiwan;
n88071053@gs.ncku.edu.tw
* Correspondence: ryyang@mail.ncku.edu.tw

**Abstract:** In 2004, the International Maritime Organization (IMO) adopted the International Convention for the Control and Management of Ships Ballast Water and Sediments (BWM Convention). Taiwan's government has been in line with the BWM Convention's obligations by passing several administrative orders and adopted the 3 + 1 Port State Control (PSC) procedure. International trade ports in Taiwan include commercial ports and exclusive industrial harbors. The industrial harbor in Taiwan is unique in the world, so the 3 + 1 PSC procedure cannot be directly applied to the industrial harbor. Based on document analysis, this study discusses the similarities and differences between commercial ports and industrial harbors. The regulations and systems for ballast water management in Taiwan and how they can be applied to industrial harbors are also discussed. Judging from the results of this study of regulations, commercial ports and industrial harbors differ in applicable laws, competent authorities, and construction and management units. However, in operational practice, industrial harbors should be regarded as a commercial port whose use is restricted. Therefore, this study posits that industrial harbors should be classified as commercial ports in Taiwan's ballast water management system. Classifying industrial harbors as falling outside commercial ports may cause management difficulties and may even cause trouble for international shipping. It is suggested that the Ministry of Economic Affairs (MOEA) first discusses with the Ministry of Transportation and Communications (MOTC) and the Ocean Affairs Council (OAC) to confirm whether industrial harbors are inside or outside the category of commercial ports, and then decide on a management system and suitable laws and regulations for integrating industrial harbors with commercial ports and international affairs.

**Keywords:** BWM Convention; commercial port; exclusive industrial harbor; 3 + 1 PSC procedure

## 1. Introduction

The main transport mode for global trade is ocean shipping; over 90% of traded goods are carried by ship [1]. Human activities such as commercial expansion, trade globalization, and the growth of fleet sizes have inadvertently led to the spread of aquatic non-indigenous species in the oceans [2–4]. According to the guidelines published by the International Union for Conservation of Nature (IUCN) in 2000, a non-indigenous species (NIS) means a species, subspecies, or lower taxon occurring outside of its natural range (past or present) and dispersal potential [5]. Aquatic NIS may be brought into another country during transportation by becoming attached to the ship hull or in the ballast water [6–8].

It is estimated by the IMO that some 3–5 million tons of ballast water is transferred throughout the world each year with ships [9]. Ballast water exchange is necessary for the safety of ship navigation [10], and it serves as a vector for the introduction of aquatic NIS [11]. If aquatic NIS enter new areas through ballast water and become alien invasive species, it may cause harm to ecosystems [12]. The introduction of alien invasive species takes place not only between different continents, but also in adjacent coastal areas [13,14]. Once they thrive in the new environment, the adverse effects on ecosystems, economies, and even human health may also increase over time [15,16].

Ballast water and sediment may introduce aquatic alien species that affect the marine environment and raise questions about the invasion of microbial pathogens [17]. In 2004, IMO adopted the BWM Convention, in order to prevent, minimize, and ultimately eliminate the transfer of harmful aquatic organisms and pathogens through the control and management of ships' ballast water and sediments [18,19]. The BWM Convention entered into force on 8 September 2017. Taiwan is not an IMO member and cannot sign the BWM Convention due to China's political interference. However, Taiwan's government has been in line with the BWM Convention's obligations by passing several administrative orders and adopt the 3 + 1 PSC procedure for ballast water management [20].

International trade ports in Taiwan include commercial ports and exclusive industrial harbors according to the different competent authorities. The competent authority of the commercial ports is MOTC, and the laws involved are the Commercial Port Law (CPL); the competent authority of the exclusive industrial harbor is MOEA, and the laws involved are CPL and the Statute for Industrial Innovation (SII) [21]. The exclusive industrial harbor in Taiwan is unique in the world. The 3 + 1 PSC procedure used in the commercial ports cannot be directly applied to exclusive industrial harbors due to differences in the competent authorities and the relevant laws. In order to improve the ballast water management in the exclusive industrial harbors and avoid bothering ships, it is necessary to develop a suitable ballast water management system for Taiwan's exclusive industrial harbor.

In this study, the current regulations and procedures for ballast water management in Taiwan were collected. The ballast water management system in the Mailiao exclusive industrial harbor is discussed by analyzing the regulations and enforcement results of Taiwan. It also puts forward suggestions for improving the ballast water management policy in the exclusive industrial harbor as a reference for the competent authority's future policy.

## 2. Materials and Methods

In order to understand how ballast water management systems work in the exclusive industrial harbor and consider whether further improvements can be made, this study discusses, based on document analysis, the following:

- The similarities and differences between commercial ports and exclusive industrial harbors;
- The regulations and systems for ballast water management in Taiwan; and
- How a ballast water management system can be applied to exclusive industrial harbors.

Next, the researchers conducted a questionnaire survey and interview with staff members responsible for ballast water management in the industrial harbor and collected opinions and suggestions on ballast water management in industrial harbors to understand the actual process of international ships entering the industrial harbor.

### 2.1. Document Analysis

Document analysis is as a method of systematically and objectively defining, evaluating, and comprehensively demonstrating to determine certainty and conclusions about past events. Its main purpose is to understand the past, gain insight into the present, and predict the future [22]. In this study, the legal resources on the construction and management of commercial port and exclusive industrial harbor were collected from government agencies and the Internet to compare the similarities and differences between commercial ports and exclusive industrial harbors. In addition, the current regulations, procedures, and implementation results for ballast water management in Taiwan were collected to understand the current situations of ballast water management in Taiwan's industrial harbor and examine whether the current regulations and systems can be strengthened.

### 2.2. Questionnaire Survey and Interview

Questionnaire surveys are a technique for gathering statistical information about the attributes, attitudes, or actions of a population by a structured set of questions. Administered by mail, in person, through the Internet, and over the telephone, questionnaire surveys provide broad coverage of populations enabling us to explore spatial and social

variations in people's attributes, attitudes, and actions. The aim was to obtain information suitable for statistical analysis [23]. This study focused on the 3 + 1 PSC procedure for ballast water management adopted by MOTC as the main survey content to investigate the perception of staff members at industrial harbors on ballast water management through questionnaires and interviews.

### 2.3. Introduction of the Area Studied

### 2.3.1. Introduction of Mailiao Exclusive Industrial Harbor

Currently, Taiwan has two industrial harbors in operation: Mailiao and Hoping. The Kwuntong industrial harbor is still under construction. The locations of the exclusive industrial harbors are shown in Figure 1.

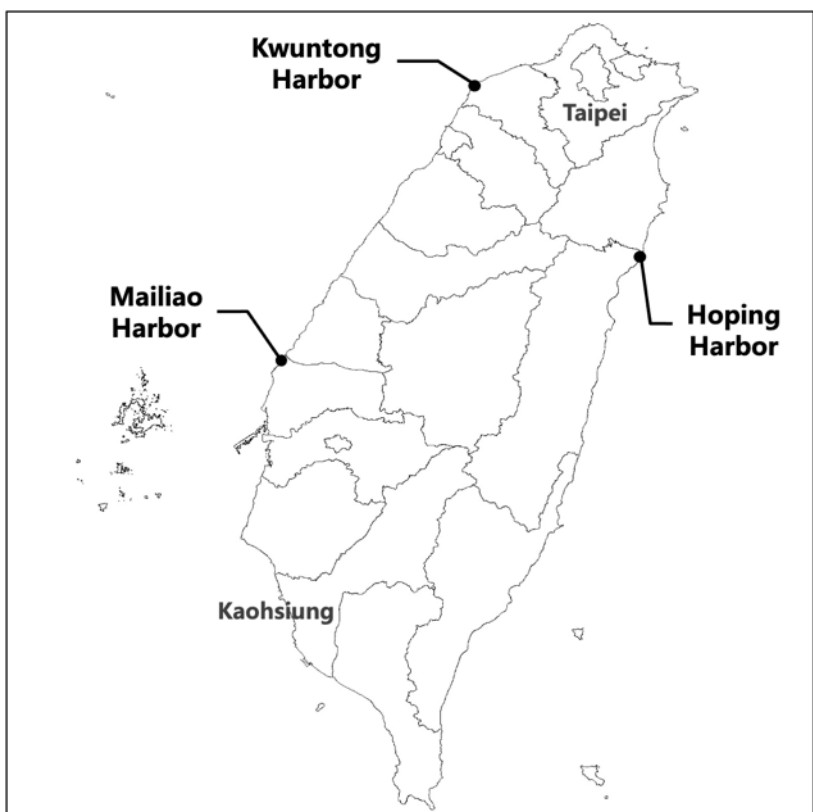

**Figure 1.** Location map of the exclusive industrial harbors in Taiwan.

Mailiao Exclusive Industrial Harbor is in the Mailiao District of the Yunlin Offshore Industrial Park. MOEA granted permission to the Mailiao Harbor Administration Co., Ltd. (MHAC) to invest in the construction of this harbor, which began operations on 1 March 2001. The harbor mainly serves the Sixth Naphtha Cracker Project. Formosa Plastics Group and Chang Chun Group are the harbor's major customers.

There are 20 port terminals in Mailiao Harbor to berthing ships for loading and unloading goods, with a throughput of 60 million tons in 2020, ranking third in Taiwan [24]. From the data of the calling at the port between 2018 and 2021, the ships into the Mailiao Harbor can be roughly divided into five origins: China, Northeast Asia, Southeast Asia, the Middle East, and Other (see Figure 2). Ships from China account for the largest share (49.7%), followed by Northeast Asia (21.5%). The first port of call for most ships departing from Mailiao Harbor is China. Because the Mailiao District is the base of the Sixth Naphtha Cracker Project, most ships calling at Mailiao harbor are oil tankers, followed by liquefaction ships. Mailiao Harbor does not handle intermodal containers now.

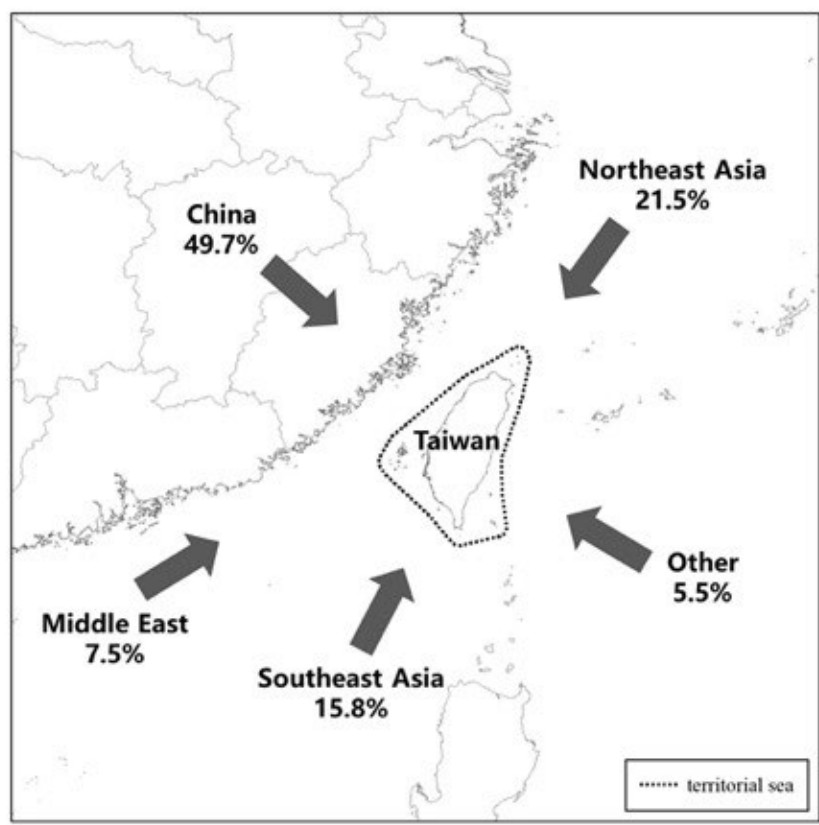

**Figure 2.** The percentage of the last port of calls originated for ships coming into Mailiao Harbor.

2.3.2. Mailiao Exclusive Industrial Harbor Ship Ballast Water Control Process

Within the framework of the BWM Convention, the Marine Pollution Control Act (MPCA), CPL, the Law of Ships (LOS), MHAC liaised with the manufacturers in the industrial park and obtained their authorization to regulate the way ships call at the harbor under a business contract. MHAC also published the "Guidelines for Controlling Ballast Water Discharge from Ships in Mailiao Exclusive Industrial Harbor" (see Appendix A), as the basis for the manufacturers in the industrial park to independently manage the ships' ballast water at Mailiao Harbor [25].

Before calling at Mailiao Harbor, international ships must submit a port entry form for verification along with a ballast water declaration form (D1) or ballast water management certificate (D2) to the Maritime and Port Bureau (MPB) in accordance with Article 64 of SII and Article 19 of CPL, and then apply to the MOEA to call in at the harbor. As the last step, the MHAC will notify the ships of its permission to enter the harbor in the assigned sequence.

After a ship has called into a harbor, MHAC may pay an unscheduled inspection visit to check the discharge records of the ship's ballast water. If there is any doubt concerning the discharge record, the MHAC will notify the MPB, which may conduct PSC inspections. From 2018 to 2021, 9839 ships called into the harbor (including 2159 domestic transshipment ships), and the MHAC conducted 1690 ship inspections, whose results all met the ballast water control standards of Mailiao Harbor, and no ships were reported as notified to the MPB for further official inspection.

In addition, in accordance with the MOTC Circular No. 0910012771 of 20 December 2002 and Articles 58–60 of CPL, foreign merchant ships call into Mailiao Harbor under MPB's control as per the regulations promulgated by IMO. From 2018 to 2021, 62 ships were subject to PSC inspections by the Central Taiwan Maritime Affairs Center, MPB, and there has been no record of ships discharging ballast water in violation of the regulations. The ship's arrival and departure procedure for ballast water management in Mailiao Harbor is shown in Figure 3, and the statistics of inspection results from 2018 to 2021 are shown in Table 1.

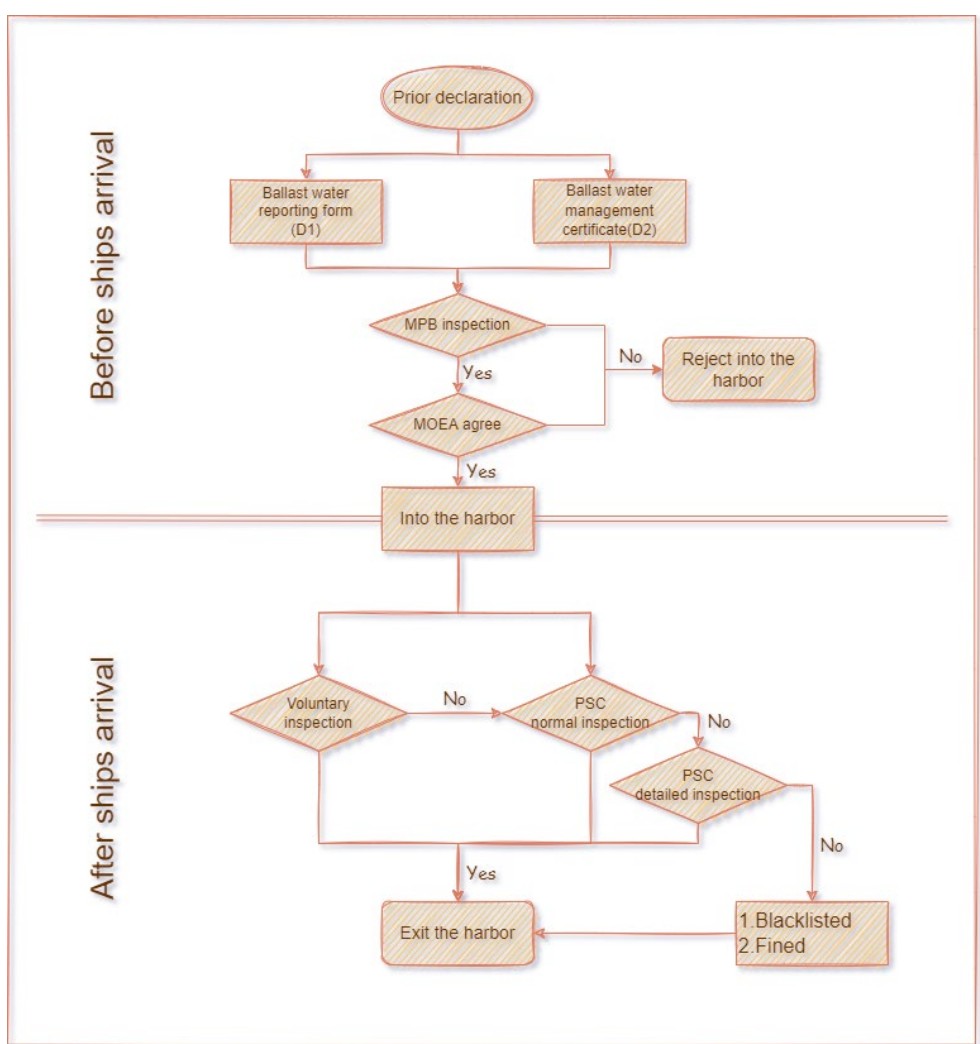

**Figure 3.** Flow chart of ballast water management into the Mailiao Industrial Harbor (Yes: Approve or agree; No: Reject or disagree).

**Table 1.** Ship Inspection Statistics of Mailiao Industrial Harbor.

| Year | Number of Ships | Voluntary Inspections of the Industrial Harbor | PSC Inspections | Blacklist of Ship Violations | Number of Fines |
|------|-----------------|-----------------------------------------------|-----------------|------------------------------|-----------------|
| 2018 | 2680 | 476 | 9 | 0 | 0 |
| 2019 | 2634 | 682 | 23 | 0 | 0 |
| 2020 | 2273 | 304 | 16 | 0 | 0 |
| 2021 | 2252 | 228 | 14 | 0 | 0 |
| Total | 9839 | 1690 | 62 | 0 | 0 |

## 3. Results and Discussion

### 3.1. Similarities and Differences between Commercial Ports and Industrial Harbors in Taiwan

The legislative background of the exclusive industrial harbor at the time was that the enterprises in the industrial park had a real need for dedicated docks, but the MOTC could not assort. After the Industrial Development Bureau, MOEA (IDB) coordinated with the MOTC, and the Statute for Upgrading Industries (now SII) was enacted in 1990. In this statute, the provisions on industrial harbors provide the competent authority in charge

of industry with the legal basis for setting up industrial harbors in industrial parks after consultation with and approval from the MOTC, and allows enterprises in such parks to lease land and build a harbor themselves. This study summarizes and analyzes the legal sources, competent authorities, and construction and management organizations of commercial ports and industrial harbors as follows:

### 3.1.1. Commercial Ports

Commercial ports are handled in accordance with the CPL. Its competent authority is the MOTC, and MPB and the Taiwan International Ports Corporation (TIPC) are responsible for the operations and management of the international commercial ports. MPB is responsible for policy concerning Taiwan's international ports (maritime and port public administration), while TIPC is responsible for those ports' operations (see Figure 4).

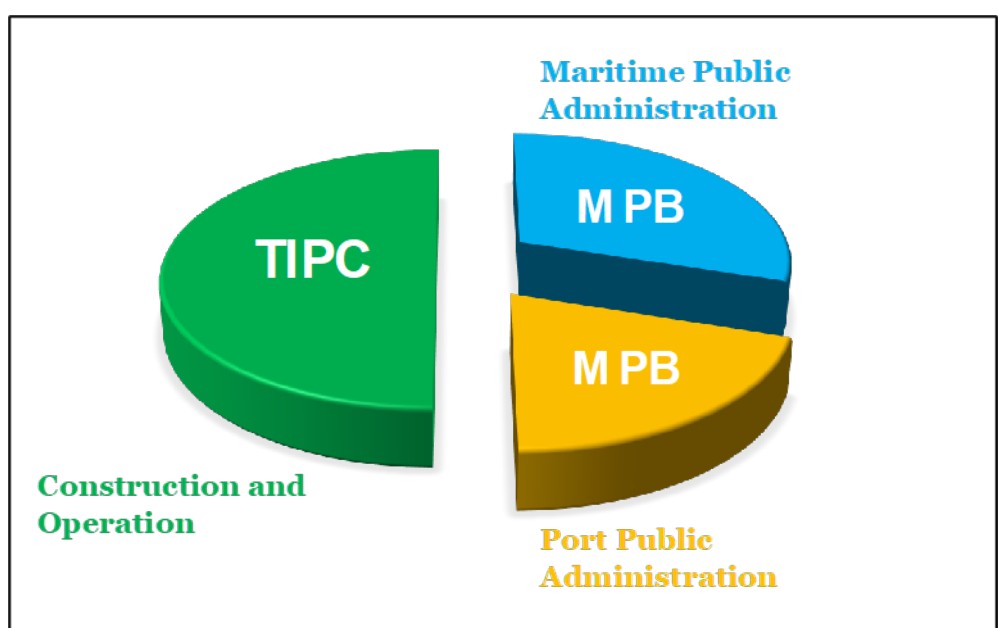

**Figure 4.** Division of authority and responsibility of commercial ports.

- Article 4 of CPL lays out the procedures for the designation of commercial ports and the delineation of port areas as follows:
  1. For the designation of an international commercial port, the MOTC shall report to the Executive Yuan (EY) for approval and promulgation.
  2. The MOTC must consult with the Ministry of the Interior (MOI) and appropriate authorities to delineate a commercial port area and the administration area before reporting to the EY for approval.
- Article 10 of CPL: Except for breakwaters, navigation channels, turning basin, navigation aids, public roads and information, gate sentry, control facilities, etc., commercial port public infrastructures of free trade zones, various facilities inside international commercial port areas are constructed and maintained by the TIPC. Other port parts may be invested in, constructed, or leased by private enterprises in an agreed manner.
- Article 44 of CPL stipulates that the MOTC formulates the Regulations on Port Services at Commercial Ports (RPSCP). Norms concerning the declaration of ship ballast water and the prohibition of discharging untreated ballast water have been revised in these regulations.
- Articles 58~60 of CPL stipulate that, according to the PSC procedures and regulations promulgated by IMO or its related agencies, MPB may inspect ship certificates, safety, equipment, and crew quotas of foreign merchants calling at and leaving Taiwan's commercial ports.

- Article 75 of the CPL stipulates that when commercial port safety and management items involve international affairs, the MOTC may refer to international conventions, agreements, and rules, methods, standards, recommendations of its supplementary rules. MOTC announced the adoption of the BWM Convention in 2015 and implemented it on 8 September 2017. This regulates that ships calling into Taiwan's commercial ports and exclusive industrial harbors on international routes must comply with the BWM Convention.

3.1.2. Exclusive Industrial Harbors

Taiwan's exclusive industrial harbors are handled in accordance with the SII (mutatis mutandis CPL) and CPL. The competent authority is the MOEA (IDB), which is responsible for the port's public administration and may approve of the investment, construction, operations, and management of ports by private enterprises; the navigation administration and PSC inspections in industrial harbors are still handled by the MPB (see Figure 5).

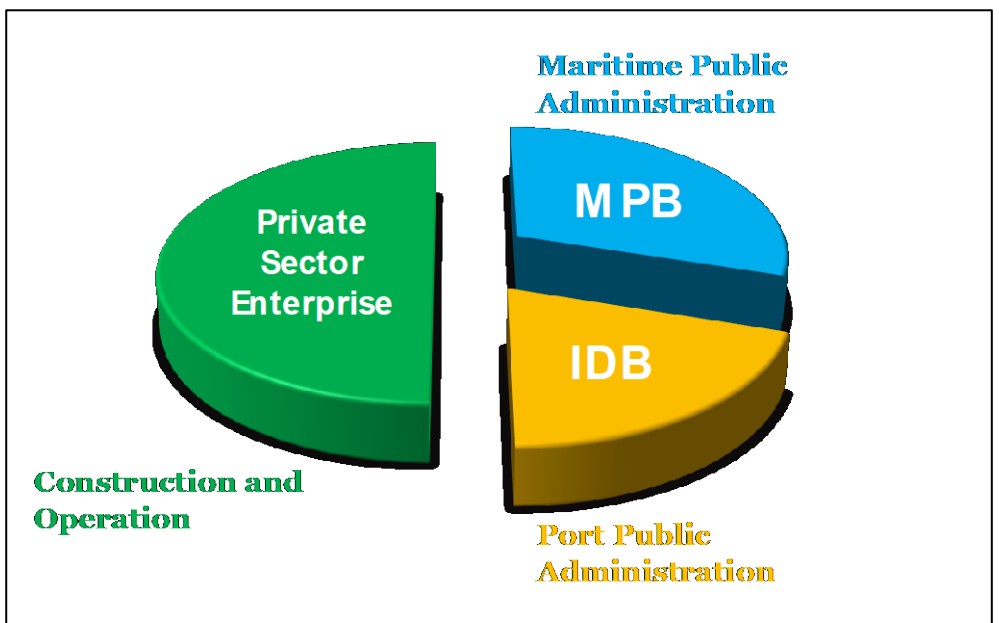

**Figure 5.** Division of authority and responsibility of exclusive industrial harbors.

- Pursuant to Article 56 of SII, the procedures for the approval and establishment of industrial harbors, harbor zone delineation, and designation are as follows:
    1. The central competent authority shall first consult with MOTC, and then submit the proposal to establish an exclusive industrial harbor to the EY for approval.
    2. The delineation of the industrial harbor area shall be submitted to the EY for approval by the MOEA after consultation with the MOTC, MOI, and other relevant agencies.
    3. For the designation of an exclusive industrial harbor, the MOEA and MOTC will, in accordance with Article 4 of CPL, submit a proposal to the EY for approval and promulgation.
- Paragraph 3 of Article 57 of the SII stipulates that an exclusive industrial harbor shall not be used for any purposes other than as an industrial park.
- Paragraph 1 of Article 58 of the SII stipulates that all facilities in industrial harbors can be invested, constructed, operated, and managed by private enterprises.
- Paragraph 5 of Article 58 of the SII stipulates that the MOEA and MOTC formulate management measures for the planning, construction, operation, and management of exclusive industrial harbors.

- Articles 58~60 of the CPL stipulates that, according to the PSC procedures and regulations promulgated by IMO or its related agencies, the port authority may inspect ship certificates, safety, equipment, crew quotas, and other matters.
- Article 64 of the SII stipulates that the planning, construction, management, operation, and safety of exclusive industrial harbors are subject to the CPL.
- Article 64 of the SII applies to Article 50 (currently Article 75) of the CPL. For commercial port safety and management matters involving international affairs, the MOEA may refer to the rules, measures, and standards stipulated in international conventions or agreements and their annexed rules, recommendations, and procedures, and adopt and implement those.

The most unique point of the industrial harbor in Taiwan is the special purpose restriction in Paragraph 3 of Article 57 of the SII. The industrial harbor shall not be used for any purposes other than as an industrial park, which greatly restricts the development of the industrial port and makes Taiwan's industrial harbor unique in the world. Only ships that are carrying goods in the industrial park can call into the industrial harbor. In addition, the industrial harbor currently does not operate passenger ships. These reduce the need for manpower, with industrial harbors operating with less manpower than commercial ports.

3.1.3. Intermediate Summary

Judging from the results of this study of regulations (see Table 2), commercial ports and industrial harbors differ in applicable laws, competent authorities, and construction and management units. However, from the perspective of historical background and legal design, the planning, construction, management, operation, and safety of industrial harbors are also partially handled according to the CPL. In addition to the stipulations in the SII, the establishment of industrial harbors and various regulations must all be discussed and formulated by the MOEA in conjunction with the MOTC before being promulgated. Therefore, in operational practice, an industrial harbor should be regarded as an international commercial port whose use is restricted.

**Table 2.** Summary of differences between commercial ports and exclusive industrial harbors.

| Item | Commercial Port | Exclusive Industrial Harbor |
| --- | --- | --- |
| Legal basis | CPL | SII |
| Competent authority | MOTC | MOEA |
| The appointment of an international trade port | MOTC reports to EY | MOEA and MOTC jointly report to EY |
| The permitted users | No restrictions | For industrial park only |
| Private enterprise participation | Except for control facilities | No restrictions |
| PSC | MPB | MPB |
| International affairs | MOTC | MOEA |

*3.2. Review of the Current Status of Ballast Water Management Regulations and Systems in Taiwan*

Due to the political conflict with China, Taiwan is not an IMO member and cannot sign the BWM Convention. However, Taiwan currently uses its MPCA, CPL, and Law on Ships to align its management regulations and systems with the convention in terms of regulating ballast water when international ships call into and leave from Taiwan's territorial seas. If international merchant ships in Taiwan's territorial seas are found to be illegally discharging untreated ballast water, they will be penalized by the MOTC or OAC, depending on whether the discharge location is inside or outside the commercial port. The following is a summary and analysis of the actions of the MOTC and OAC in response to the BWM Convention:

### 3.2.1. MOTC (CPL, LOS)

MOTC is the competent authority of the CPL and LOS. The scope of application of the CPL is the commercial port areas of Taiwan, so the management of ballast water outside the port area is not within its jurisdiction. Thee LOS governs ships of the Republic of China and does not apply to foreign merchant ships. In addition to adopting the BWM Convention under the CPL and LOS, the MOTC has also revised the RPSCP and Regulations on Equipment of Ships (RES). The MOTC has published the following announcements and regulations on ballast water:

- On 20 August 2015, Jiao-Hang-Zi Order No. 10498001451 announced the adoption of the BWM Convention in accordance with Article 75 of the CPL and Article 101 of the LOS to regulate the exchange and discharge of ballast water for calling into Taiwan's commercial ports and industrial harbors on international routes in accordance with the BWM Convention. The order was implemented on 8 September 2017 (applicable to commercial ports and industrial harbors).
- On 5 October 2015, Jiao-Hang-Zi Order No. 10450127681 was added to Article 224–1 of the RES, adding that ships on international routes must be equipped with a ship ballast water management system. The order applies recognition rules similar to the BWM Convention and was implemented on 8 September 2017 (applicable to Taiwan ships, not applicable to foreign ships).
- On 6 October 2015, Jiao-Hang-Zi Order No. 10450128691 was amended and promulgated Articles 3 and 20 of the RPSCP: Ships on international routes calling at Taiwanese ports must submit a ballast water forecast form and are prohibited from the discharge of untreated ballast water in the port area. This order was implemented on 8 September 2017 (applicable to commercial ports, not applicable to industrial harbors).

### 3.2.2. OAC (MPCA)

OAC is the competent authority of the MPCA (the Environmental Protection Agency; the Executive Yuan (EPA) had this role before 2018). The scope of application of the MPCA covers the intertidal zones, internal waters, territorial waters, adjacent areas, exclusive economic zones, and continental reefs under the jurisdiction of Taiwan (including commercial ports and industrial harbors). The EPA has banned the exchange of ballast water and the discharge of untreated ballast water in Taiwan's territorial seas according to the MPCA. The summary is as follows:

- EPA Announcement No. 1050005934 of 26 January 2016 stipulates that the untreated ballast water of ships falls under Article 3, Paragraph 6 of the MPCA. According to Article 29, Paragraph 1 of the same law, untreated ballast water of ships must be retained inside the ships or discharged to on-shore reception facilities. This announcement was also implemented on 8 September 2017 (applicable to Taiwan's territorial seas, commercial ports, and industrial harbors).
- EPA Announcement No. 1050005934A of 26 January 2016 stipulates that the exchange of ballast water in Taiwan's territorial seas is prohibited. This announcement was also implemented on 8 September 2017 (applicable to Taiwan's territorial seas, commercial ports, and industrial harbors).

### 3.2.3. Taiwan's Ballast Water Management System Comply with the BWM Convention

After the Taiwan government's amendments to regulations and public announcements, the entry of international merchant ships into Taiwan's territorial seas is not only handled in accordance with the BWM Convention, but is also regulated by the MPCA and CPL. Inside and outside the commercial ports, the MOTC and OAC each control the discharge of ballast water in accordance with the CPL and MPCA, and the principle and administrative principles of the special law should be adopted before the common law, which will force ships to keep untreated ballast water on board or discharge it onshore to collection facilities to ensure the ecological integrity of Taiwan's seas.

Taiwan currently implements the so-called 3 + 1 hierarchical PSC procedure to implement the BWM Convention. The procedure was developed and amended by the MOTC, OAC, EPA, the Council of Agriculture (COA), and the Ministry of Health and Welfare (MOHW). On 29 June 2015, relevant units were instructed to handle matters accordingly through Letter Chuan-Bo-Zi No. 1040057286. The current management mechanism is shown in Table 3.

**Table 3.** Ballast water management: The 3 + 1 hierarchical PSC procedure in Taiwan.

| Phase | Work Content | Implementers |
|---|---|---|
| Voluntary management of ship | 1. All ships are required to implement a ballast water and sediment management plan. <br> 2. All ships will also have to carry a ballast water record book and an international ballast water management certificate. <br> 3. Shipping companies annually verify the ballast water management of their ships in accordance with the International Safety Management Code (ISM Code). | Shipping companies/ Ship owners/ captains |
| First level | 1. All ballast water records of ship are spot-checked by the MPB. <br> 2. Ships with doubts about ballast water records will be inspected by PSC officers, checking the water quality if necessary. <br> 3. If a violation is found, the ship shall be penalized by law. <br> 4. Maintaining a blacklist of ship violations. | MOTC |
| Second level | 1. Strengthening the inspection for blacklisted ships by PSC officers. <br> 2. If a violation is found, the ship shall be penalized by law. | MOTC |
| | 1. The ballast water discharge of blacklisted ships will be monitored by the Coast Guard Administration, OAC when it enters the territorial seas of Taiwan. <br> 2. If a violation is found, the ship shall be penalized by law. | OAC |
| Third level | 1. For a ship that is on the blacklist or has ever called into an epidemic area, ballast water sampling test will be mandatory. <br> 2. Ships failing to meet the water quality standards shall be penalized. <br> 3. Assessing the level of risk for potential areas of introduction and exploring measures of improvement. | MOTC/ OAC/ COA / MOHW |

Sources: MPB

*3.3. Questionnaire Survey and Interview on Ballast Water Control Measures in Mailiao Exclusive Industrial Harbor*

For this study, surveys and interviews were conducted with the concerned authorities, harbor management agencies, shipping companies, and shipping agencies by means of telephone or Internet communication software and electronic questionnaires to collect and understand opinions and suggestions on ballast water management in industrial harbors. Since Mailiao Harbor is used by the 6th Naphta Cracker Project, almost all shipping and cargo docking and unloading operations are handled by subsidiaries of the Formosa Plastics Group. The respondents of this study are sufficient to represent the situation in Mailiao Harbor. The questionnaires are shown in Appendix B, and the interviews and survey objects are shown in Table 4.

**Table 4.** The list of questionnaire and interview objects.

| Category | Name | Job Tenure | Job Title |
|---|---|---|---|
| Government sector | IDB | 8 | Technical specialist |
| | | 4 | Technical specialist |
| | | 13 | Officer |
| | | 22 | Officer |
| | | 36 | Senior manager |
| Harbor management institution | MHAC | 30 | Assistant Vice President |
| | | 26 | Department Manager |
| | | 25 | Advanced Administrator |
| Vessel Carrier | Formosa Plastics Marine Corporation | 41 | Department Manager |
| | | 14 | Advanced Administrator |
| Shipping Agency | Formosa Plastics Navigation Corporation | 25 | Department Manager |
| | | 15 | Advanced Administrator |
| | | 15 | Advanced Administrator |
| | | 16 | Advanced Administrator |

The interview dates were from October 2021 to February 2022. Four questions were formulated for this study to gauge the level of understanding of the industrial harbor management authorities/organizations and operators on the ballast water management system in Mailiao Harbor. The results of the questionnaire interview in Table 5 show that almost all colleagues working in the industrial harbor were aware of Taiwan's 3 + 1 ballast water management procedure (except for IDB new staff). In addition, three staff members from government departments have put forward suggestions for the management system of ship ballast water in Mailiao Harbor, and these suggestions will be discussed in the next section.

**Table 5.** Questionnaire results.

| Category | Q1 | | Q2 | | Q3 | | Q4 | |
|---|---|---|---|---|---|---|---|---|
| | Yes | No | Yes | No | Yes | No | Yes | No |
| Government sector | 4 | 1 | 4 | 1 | 5 | 0 | 3 | 2 |
| | 80% | 20% | 80% | 20% | 100% | 0% | 60% | 40% |
| Harbor management institution | 3 | 0 | 3 | 0 | 3 | 0 | 0 | 3 |
| | 100% | 0% | 100% | 0% | 100% | 0% | 0% | 100% |
| Vessel Carrier | 2 | 0 | 2 | 0 | 2 | 0 | 0 | 2 |
| | 100% | 0% | 100% | 0% | 100% | 0% | 0% | 100% |
| Shipping Agency | 4 | 0 | 4 | 0 | 4 | 0 | 0 | 4 |
| | 100% | 0% | 100% | 0% | 100% | 0% | 0% | 100% |
| Total | 13 | 1 | 13 | 1 | 14 | 0 | 3 | 11 |
| | 93% | 7% | 93% | 7% | 100% | 0% | 21% | 79% |

- It is recommended to clarify whether industrial harbors are commercial ports or waters outside commercial ports to participate in seizures as a form of penalization for violating ships (CPL or MPCA).
- It is recommended that a dedicated unit is set up and a special law (Ballast Water Management Regulation) is formulated for comprehensive management.
- IDB should invite relevant units to discuss and clarify the applicability of laws and regulations and management topics related to industrial harbors.

*3.4. The Application of Taiwan's Ballast Water Management Regulations and Systems to Industrial Harbors*

Industrial harbors are a port type unique to Taiwan. As above-mentioned in Section 3.1, we know that commercial ports and industrial harbors differ in applicable laws, competent authorities, and construction and management units. The 3 + 1 hierarchical PSC procedure for commercial ports cannot be directly applied to industrial harbors. For instance, the RPSCP cannot be applied to industrial harbors (see Table 6). If industrial harbors are recognized as outside commercial ports, the OAC must follow the MPCA to manage them. Moreover, other regulatory issues do not align with operational practice, which may likely cause trouble for international shipping. Since the management units in Table 3 do not list the MOEA, this section will clarify and discuss the issues of regulations, systems, and authority for the management of ship ballast water in industrial harbors:

**Table 6.** Taiwan ballast water authority and regulations.

| Administrative orders NO. | Contents | Commercial Port | Exclusive Industrial Harbor |
|---|---|:---:|:---:|
| No Jiao-Hang-Zi 10498001451 | MPB announced that the measures of the BWC Convention would be adopted in Taiwan. | ○ | △ |
| No Jiao-Hang-Zi 10450127681 | The amended RES added:<br>1. All ships shall install ballast water management systems.<br>2. Ballast water management systems must be approved by the Administration in accordance with IMO guidelines. | ○ | ○ |
| No Jiao-Hang-Zi 10450128691 | The amended RPSCP added:<br>1. The mandatory declaration of ballast water before ships arrives at commercial ports in Taiwan.<br>2. Discharging of untreated ballast water in the port area is also prohibited. | ○ | ✕ |
| No Huan-Shui-Zi 1050005934 | The untreated ballast water of the vessels is identified as a pollutant in the MPCA. | ○ | ○ |
| No Huan-Shui-Zi 1050005934A | Taiwan's territorial sea is marine control area where ballast water exchange is prohibited. | ○ | ○ |

○: Applicable △: Mutatis mutandis ✕: Not applicable

3.4.1. In Taiwan's Ballast Water Management System, Should an Industrial Harbor Be Identified as a Commercial Port or Waters Outside Commercial Ports?

As mentioned in Section 3.2, Taiwan's ballast water management is divided between outside or within commercial ports. The MOTC and OAC control the discharge of ballast water in accordance with CPL and MPCA. Therefore, it is necessary to clarify whether industrial harbors should be identified as commercial ports or be managed outside that scope. As mentioned in Section 3.1, industrial harbors are to be regarded as restricted international commercial ports when it comes to operational practice. Therefore, this study posits that industrial harbors should be classified as commercial ports in Taiwan's ballast water management system, in line with the current legal system and operational practice. Classifying industrial harbors outside that scope may cause management difficulties and may even hinder international shipping.

### 3.4.2. In Taiwan's Ballast Water Management System, Is There a Dedicated Unit and a Special Law to Deal with It? What Is MOEA's Role or Task?

The regulation of Taiwan's ballast water management by a special law formulated by a dedicated unit was denied when the EY discussed the current ballast water management system in 2015. This study will not revisit that discussion, but solely focus on analyzing ballast water management in industrial harbors. The MOEA is the competent authority in charge of industrial harbors. Unless it is determined that industrial harbors are waters outside commercial ports and must be managed by the OAC in accordance with the MPCA, pollution cases in industrial harbors must be handled by the MOEA according to the SII with regard to Taiwan's ballast water management system. With a view to current operational practice in industrial harbors, this study posits that MOEA should assist in PSC inspections and act as the dedicated management unit of post-violation penalization to effectively manage the ballast water of ships in industrial harbors and protect Taiwan's maritime ecology.

- PSC inspections in industrial harbors are currently carried out by the MPB, who is assisted by the harbor management institution. Therefore, the MOEA can be listed as a supporting unit for PSC inspections in Taiwan's ballast water management system, responsible for assisting the MPB with PSC inspections and amendments in law.
- According to special law precedence over the common law principle and administrative division of labor, an incident of a ship causing pollution in an industrial harbor should be penalized in accordance with the SII by the MOEA. Therefore, the MOEA must penalize violations in industrial harbor under Taiwan's ballast water management system. As the competent authority, it will be responsible for the penalties for violations found in PSC inspections. In addition, Table 1 shows that no ships have ever been found to be in violation of these regulations, so the MOEA, MOTC, and OAC should first discuss which regulations should be applied to violations of ballast water management regulations in industrial harbors.

### 4. Conclusions

This study has collected the current Taiwan ballast water management system and practices from Taiwan's government agencies. Therefore, we used Mailiao Harbor as an example to clarify and discuss the current situation and outstanding issues in the ship entry declaration system and the legal sources of violation investigation and penalization in industrial harbors.

As mentioned in Section 3.4, the 3 + 1 hierarchical PSC procedure for commercial ports cannot be directly applied to industrial harbors, and the management units in Table 3 do not list the MOEA. However, pollution cases in industrial harbors must be handled by the MOEA according to the SII. Therefore, it is necessary to revise a suitable ballast water management system for Taiwan's exclusive industrial harbor.

This study argues that industrial harbors should be classified as commercial ports in Taiwan's ballast water management system, in line with the current legal system and operational practice. Classifying industrial harbors as falling outside commercial ports may cause management difficulties and may even cause trouble for international shipping. It is suggested that the MOEA first discusses with the MOTC and OAC regarding the ballast water management system in industrial harbors to confirm whether industrial harbors are inside or outside the category of commercial ports, and then decide on a management system and suitable laws to integrate industrial harbors with commercial ports and international affairs. Thee SII is to be revised when necessary. Suggestions on the classification of industrial harbors outside or within commercial port areas are given below for the reference of competent authorities in their decision-making.

*4.1. Classifying Industrial Harbors as Commercial Ports*

If it is determined that industrial harbors are commercial port areas in Taiwan's ballast water management system, it is suggested that the MOEA consider revising the SII and its sub-laws and other supporting methods, and coordinate with the MPB to revise the current management system. The suggested actions are as follows:

- The MOTC announces that the BWM Convention also covers industrial harbors. Afterward, international conventions should be adopted in accordance with Article 75 of the CPL:
    1. When commercial ports and industrial harbors as a whole fall under international conventions, the MOTC and MOEA should announce it jointly.
    2. When commercial ports and industrial harbors do not as a whole fall under international conventions, the MOTC and MOEA should announce the applicable items separately.
- Ships calling into industrial harbors must report their ballast water management records to the MPB. In the event of doubt, PSC inspections and random water quality inspections must be conducted as appropriate. If records are falsified or the water quality is inconsistent, they will be included in the MPB's ship ballast water blacklist.
- When MPB staff enter an industrial harbor for PSC inspections, they may request the management agency of the industrial harbor to accompany and assist them, and the management agency of the commercial port cannot refuse.
- If untreated ballast water is found in an industrial harbor, the MOEA should, after law amendments, penalize the violator under the SII (for example, Article 66, Paragraph 3 of CPL, a fine between TWD 100,000 and TWD 500,000).

*4.2. Classifying Industrial Harbors as Outside the Commercial Port Areas*

If it is determined that industrial harbors fall outside the commercial port areas in Taiwan's ballast water management system (i.e., located in the territorial seas), the MPCA will apply, and thee OAC will be the competent authority. Ships calling into industrial harbors do not need to declare ballast water management records to the MPB. The ship's ballast water is inspected by the OAC or the local environmental protection department. If untreated ballast water is found and discharged, the OAC or the local environmental protection department will be responsible for the inspection of the ship's ballast water. According to Article 53 of the MPCA, a fine between TWD 300,000 and TWD 1.5 million will be imposed. However, whether industrial harbors can be classified as independently falling outside the commercial port area for the ballast water management system remains to be discussed. The MOEA, MOTC, and OAC will need to discuss the ballast water management system and the related regulations for industrial harbors, and then decide on a management system and suitable laws for integrating industrial harbors with commercial ports and international affairs.

**Author Contributions:** Conceptualization, H.-N.H. and R.-Y.Y.; Methodology, H.-N.H.; Investigation, H.-N.H.; Resources, H.-N.H. and R.-Y.Y.; Data curation, H.-N.H.; Writing—original draft preparation, H.-N.H.; Writing—review and editing, R.-Y.Y.; Visualization, H.-N.H.; Supervision, R.-Y.Y.; Project administration, H.-N.H.; Funding acquisition, R.-Y.Y. All authors have read and agreed to the published version of the manuscript.

**Funding:** This research received no external funding.

**Institutional Review Board Statement:** Not applicable.

**Informed Consent Statement:** Not applicable.

**Data Availability Statement:** Not applicable.

**Acknowledgments:** This article was made possible with help from the MPB, MOTC, IDB, MOEA, and MHAC.

**Conflicts of Interest:** The authors declare no conflict of interest.

## Appendix A. Guidelines for Controlling Ballast Water Discharge from Ships in Mailiao Exclusive Industrial Harbor (The First Revision on 12 September 2017)

Article 1 Basis:

1. In 2004, the International Maritime Organization (IMO) adopted the "International Convention on the Management of Ships' Ballast Water and Sediments".
2. Paragraph 6 of Article 3 of the MPCA "Ship ballast water is the emission as specified in Paragraph 6 of Article 3 of the Marine Pollution Control Law".
3. Items 2 and 3 of Article 8 of the MPCA "The marine control area and its pollution control measures are prohibited for the exchange of ship ballast water within the territorial waters of our country".
4. Ballast water declaration form in Article 3 of the Commercial and Port Administration Regulations, and the prohibition of discharging untreated ballast water in Paragraph 8 of Article 20.
5. Ship Equipment Regulations: Chapter 3 Provision of Ship Pollution Prevention Equipment; Section 7 Ship Ballast Water Management System; and Article 224–1 regulate the setting standards of ship ballast water management systems.

Article 2: The control operations for the discharge of ballast water from ships are as follows:

1. Online forecast before entering the port: The shipping business operator must carry out the ballast water declaration operation in accordance with the principle of ship ballast water exchange and the installation of ballast water equipment to the MTNet Port Bureau of the Ministry of Communications for ballast water inbound forecasts and declaration operations. Those who do not declare will not be able to obtain an entry permit.

    (1) The exchange of ballast water on ships must be outside the territorial sea of the country and in accordance with the regulations of Article D-1 of the Convention: the exchange rate of 95% of the contained ballast water; or replacement of three times the volume of ballast water (see Annex 1).

    (2) The ship shall be equipped with ballast water equipment, and the ballast water discharged shall comply with the regulations of Article D-2 of the Convention: the standards for the type, size, and concentration of organisms contained in the ballast water (see Annex 1).

    (3) For the declaration of ballast water, the regulations of Article D-1 of the Convention or the regulations of Article D-2 of the Convention shall be adopted, and the gradual implementation schedule of the ship's ballast water management standards shall be referred to (such as Annex 1). ((1) New ships: new ships constructed on and after 8 September 2017 shall apply; (2) Existing ships: ships subject to renewal of certificate (IOPP) surveys on and after 8 September 2019, shall comply).

2. Inspection and control: Conduct unscheduled inspections aboard the ship to carry out the ballast water discharge inspection of the ship. If there is any doubt about the content of the discharge records, notify the port authority to carry out the port state control inspection. If the appearance of ballast water is obviously dirty, the ship is immediately asked to stop the discharge of ballast water. If the ship is unable to load the cargo after stopping the discharge of ballast water, the ship will, in accordance with the "Regulations on the Designation of Seats and Anchorages for Mailiao Industrial Harbor" No. 5 Item 6 of Article 6, notify the ship owner to move out of the port, and list the ship as a controlled ship according to the ship control operations, and it is forbidden to rent it without permission.

Article 3: The content of the control operations in Article 2 of these Guidelines shall be implemented in accordance with the announcement of the Ministry of Communications, and the same shall apply to revisions.

### Appendix B. Questionnaire on Ballast Water Management System for Exclusive Industrial Harbors

The MOTC announced the adoption of the international convention for the control and management of ships' ballast water and sediments in 2015, which came into effect on 8 September 2017. The conventions and management systems are also applicable to exclusive industrial harbors. The IDB intends to adjust the ballast water management system in exclusive industrial harbors with reference to the commercial ports and take opinions from all walks of life as a reference for future revisions. Thank you for taking the time to help fill out this questionnaire.

If you have any questions, please contact Horng Hao Nan, Tel.: +2-2754-1255 (ext. 2514). Please return this questionnaire by email before 28 February 2022: hnhorng@moeaidb.gov.tw

A. Identity: ☐Government sector ☐Harbor management institution ☐Vessel Carrier ☐Shipping Agency ☐Other _________

B. Institution name: _________

C. Job title: _________

D. Job tenure: _________

1. Do you know that the Ministry of Communications has announced the adoption of the "International Convention on the Management of Ships' Ballast Water and Sediments" in 2015, and it has come into effect on 8 September 2017?

   ☐Yes ☐No;
   If the answer is yes, when will you be compliant? Where did you find out?

   ______________________

2. Are you aware that merchant ships entering the Taiwan International Commercial Port must declare the ship's ballast water in advance, and that untreated ballast water cannot be discharged in the port?

   ☐Yes ☐No;
   If the answer is yes, when will you be compliant? Where did you find out?

   ______________________

3. Do you know that merchant ships entering Mailiao Industrial Harbor are also required to declare the ship's ballast water in advance, and untreated ballast water is not allowed to be discharged in the port?

   ☐Yes ☐No;
   If the answer is yes, when will you be compliant? Where did you find out?

   ______________________

4. Do you have any views or suggestions on the ship's ballast water management system in Mailiao Industrial Harbor?

   ☐Yes ☐No;

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
