# Peer review of "The Study on the Ballast Water Management of Mailiao Exclusive Industrial Harbor in Taiwan"

_water, doi:10.3390/w14091431_

Round 1

Reviewer 1 Report

The manuscript (ms) „The Study on the Ballast Water Management of Mailiao Exclu-2 sive Industrial Harbor in Taiwan“ is interesting for reader involved with maritime interest, and as such is worth publication.

There are some issues which need to be resolved before publication.

Abstract:

Too long and it should be shortened. Also, many sentences are copied directly from the rest of the ms, and should be rewritten to keep the ms interesting.

Line 12-13           International commercial ports in Taiwan include commercial ports and 12 exclusive industrial harbors according to the different competent authorities.

Is this correct? Namely it is said that International commercial ports include commercial ports.

Line 13-14 The exclusive indus-13 trial harbor in Taiwan is unique to the world.

In ms an explanation of this is needed. What makes it unique?

Line 14-19           Too long sentence. Divide in 2 or 3.

Line 23-27           Repetition.

Introduction:

Line 74-76           Sentence is not complete. Finish the sentence.

Materials and Methods:

Line 85-89           Too long. Divide in 2 or 3.

Line 120               Is it Mailiao or Mai Liao (in Fig)? Use one name in entire ms.

Line 121-122      Both harbors are shown.

Line 123-124      No verb in sentence. Correct the sentence.

Line 128-129      Is berth also a verb? Check and correct. Also, entire sentence is unclear. Correct.

Line 129-131      Add “from” or similar as now sentence is incomplete.

Line 132-133      Where from do ships leave. Add.

Line 133-135      Because and thus cannot be used in same sentence. Correct.

Line 149-170      Is this procedure presented at fig. 3? Add this information if correct.

Line 166               Use PSC in entire ms, and add full name when first used. Strange to have PSC in Abstract and full name in rest of ms.

Fig. 3. What does yes/no stand for? Add explanation in caption.

Results and Discussion:

Fig. 4 and 5         Check if MPB are used correctly.

Line 342               Double in. Erase.

Line 342-343      The questionnaires are not shown in App. A. There is only one, not more of them.

Line 348-359      This is repetition as it is shown also in App B. Remove one as not needed to have it twice in ms.

Table 4.                Did only 14 persons answer this questionnaire? What is the purpose of it?

Table 6.                What do circle, triangle and x stand for? Add explanation.

Conclusions:

It is not really clear how are Results and Disscussion related to Conclusions. Rewrite in order to relate these two segments better.

App. A.                 This material is not cited in ms.

References:

Add full list of authors instead of using et al.

Reviewer 2 Report

The paper clearly presents and discusses the topic and presents understandable conclusions. Still, I suggest only a few minor changes or alterations to the text as follows:

25-27 "This study argues that industrial harbors should be classified as commercial ports in Taiwan's ballast water management system, in line with the current legal system and operational practice." The sentence is repeated unnecessarily.

65 When first mentioning the acronym PSC, it would be good to state the full meaning of the term "Port State Control (PSC)", as stated for other acronyms in the text.

162 A space is required between the specific article "the" in front of the acronym "MHAC".

166 After the first mention of the term "Port State Control", the abbreviation "PSC" may be used below.

188 What is the meaning of the abbreviation TIPC?

249 Delete the hyphen (-) in the word "Article"!

313 Delete the hyphen (-) in the word "applicable"!

613, 625 Capital letter "P" in page citation?
